# An anionic phthalocyanine decreases NRAS expression by breaking down its RNA G-quadruplex

Keiko Kawauchi [1], Wataru Sugimoto[1], Takatoshi Yasui[1], Kohei Murata[1], Katsuhiko Itoh[1], Kazuki Takagi[1], Takaaki Tsuruoka [1], Kensuke Akamatsu[1], Hisae Tateishi-Karimata[2], Naoki Sugimoto[2] & Daisuke Miyoshi [1]

Aberrant activation of RAS signalling pathways contributes to aggressive phenotypes of cancer cells. The RAS-targeted therapies for cancer, therefore, have been recognised to be effective; however, current developments on targeting RAS have not advanced due to structural features of the RAS protein. Here, we show that expression of NRAS, a major isoform of RAS, can be controlled by photo-irradiation with an anionic phthalocyanine, ZnAPC, targeting *NRAS* mRNA. In vitro experiments reveal that ZnAPC binds to a G-quadruplex–forming oligonucleotide derived from the 5′-untranslated region of *NRAS* mRNA even in the presence of excess double-stranded RNA, which is abundant in cells, resulting in selective cleavage of the target RNA's G-quadruplex upon photo-irradiation. In line with these results, upon photo-irradiation, ZnAPC decreases *NRAS* mRNA and NRAS expression and thus viability of cancer cells. These results indicate that ZnAPC may be a prominent photosensitiser for a molecularly targeted photodynamic therapy for cancer.

[1] Faculty of Frontiers of Innovative Research in Science and Technology (FIRST), Konan University, Kobe 650-0047, Japan. [2] Frontier Institute for Biomolecular Engineering Research (FIBER), Konan University, Kobe 650-0047, Japan. Correspondence and requests for materials should be addressed to K.K. (email: kawauchi@center.konan-u.ac.jp) or to D.M. (email: miyoshi@center.konan-u.ac.jp)

Photodynamic therapy (PDT) is being widely recognised as a minimally invasive cancer treatment[1–4]. The photo-sensitisers used for PDT mostly consist of porphyrins and their analogues such as phthalocyanines, which possess low cytotoxicity in the dark and preferentially accumulate in tumour tissue[3–6]. Moreover, PDT typically uses light in a wavelength range of 600–800 nm to avoid interference by endogenous chromophores[3,7–9]. A photosensitiser absorbs light and subsequently relaxes to the first excited singlet state. Then, the singlet state undergoes conversion to the triplet state when it does not go back to the ground state. An electronic energy of the photosensitiser in the triplet state transfers to oxygen, resulting in formation of cytotoxic reactive oxygen species (ROS) such as singlet oxygen ($^1O_2$) and superoxide ($O_2^-$). Thus, cancer cells are killed by these photosensitisers in response to light exposure. Recently, in PDT, several photosensitisers targeting key molecules such as KRAS and Ki-67, which are associated with aggressive cancer cells, have been explored[10–13].

In cancer cells, commonly occurring missense mutations in three members of the RAS family genes (HRAS, KRAS and NRAS) result in their constitutive activation. In addition to these mutations, the overexpression and aberrant activation of receptor tyrosine kinases, such as epidermal growth factor receptor (EGFR) and hepatocyte growth factor receptor (HGFR/c-Met), lead to RAS hyperactivation. Because RAS hyperactivation promotes abnormal cell proliferation and metastasis, cancer therapies that involve drugs that target RAS are widely believed to be effective. Nonetheless, the drugs that directly target a RAS

protein could not be identified for a long time because of the structural features of the RAS proteins[14]. It is known that in cancer cells, NRAS and KRAS are often hyperactivated in comparison with HRAS and therefore NRAS and KRAS are much more important therapeutic targets in cancer when compared to HRAS[14,15]. Although several potential small molecules that target the mutant KRAS G12 protein have been reported[16–18], effective inhibitors of NRAS have yet to be identified.

G-quadruplexes are non-canonical structures of nucleic acids formed by guanine-rich oligonucleotides with four Hoogsteen-paired coplanar guanines, called a G-quartet[19]. The nucleotide sequences containing $G_{2-5}-N_{1-7}-G_{2-5}-N_{1-7}-G_{2-5}-N_{1-7}-G_{2-5}$, where G (stem) is guanine and N (loop) may be any nucleotide, can form G-quadruplexes[20]. G-quadruplex–forming sequences are found in telomeres and are enriched in the promoters and untranslated regions (UTRs) of genes, especially cancer-related genes such as NRAS, VEGF and BCL2[20–27]. Because DNA G-quadruplexes in the promoter often suppress gene expression through impairment of the initiation of transcription[28], Balasubramanian's and other groups have demonstrated that formation of G-quadruplexes in mRNAs regulates gene expression[21,27,29–31]. It has also been reported that RNA G-quadruplexes in the 5' UTR repress translation by interfering with the recruitment of the pre-initiation complex[32]. The gene expression regulated by formation of G-quadruplexes is assumed to be associated with cancer development and progression[27]. This notion has been supported by a growing body of evidence on G-quadruplex ligands as potential anticancer drugs[31,33–35]. Yet, there are no approved anticancer

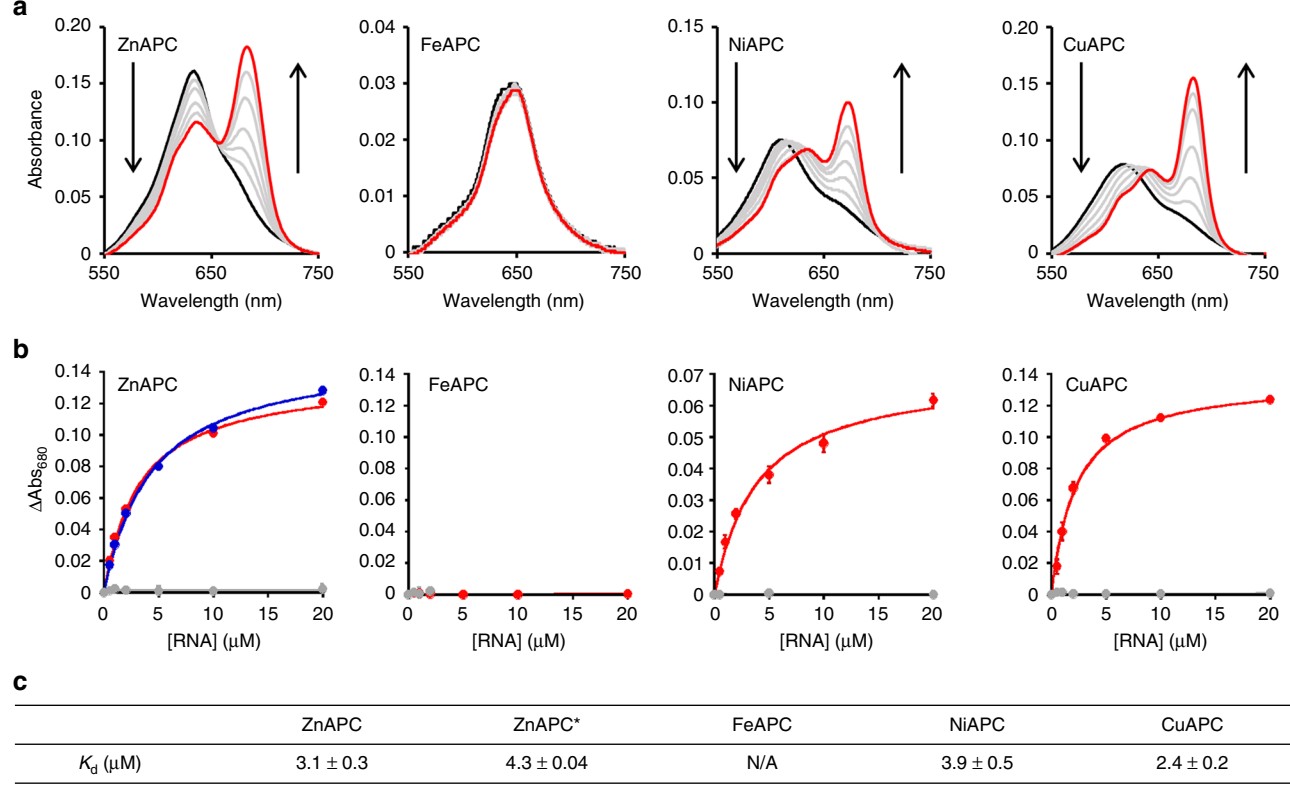

**Fig. 1** ZnAPC, NiAPC and CuAPC bind to the NRAS RNA G-quadruplex. **a** VIS absorbance spectra of 2 μM ZnAPC, FeAPC, NiAPC or CuAPC with 0, 0.5, 1, 2, 5, 10 or 20 μM NRAS RNA at 25 °C. The spectra with 0 and 20 μM NRAS RNA are highlighted in black and red, respectively. **b** Plots of ΔAbsorbance at 680 nm (=absorbance with NRAS RNA minus absorbance without NRAS RNA) of ZnAPC, FeAPC, NiAPC or CuAPC vs. the concentration of NRAS RNA (red) or dsRNA (grey). The plots of ΔAbsorbance in the presence of 100 μM dsRNA are also given in the ZnAPC spectrum (blue). Error bars represent mean ± SD; n = 3. **c** $K_d$ values of the APCs with NRAS RNA at 25 °C

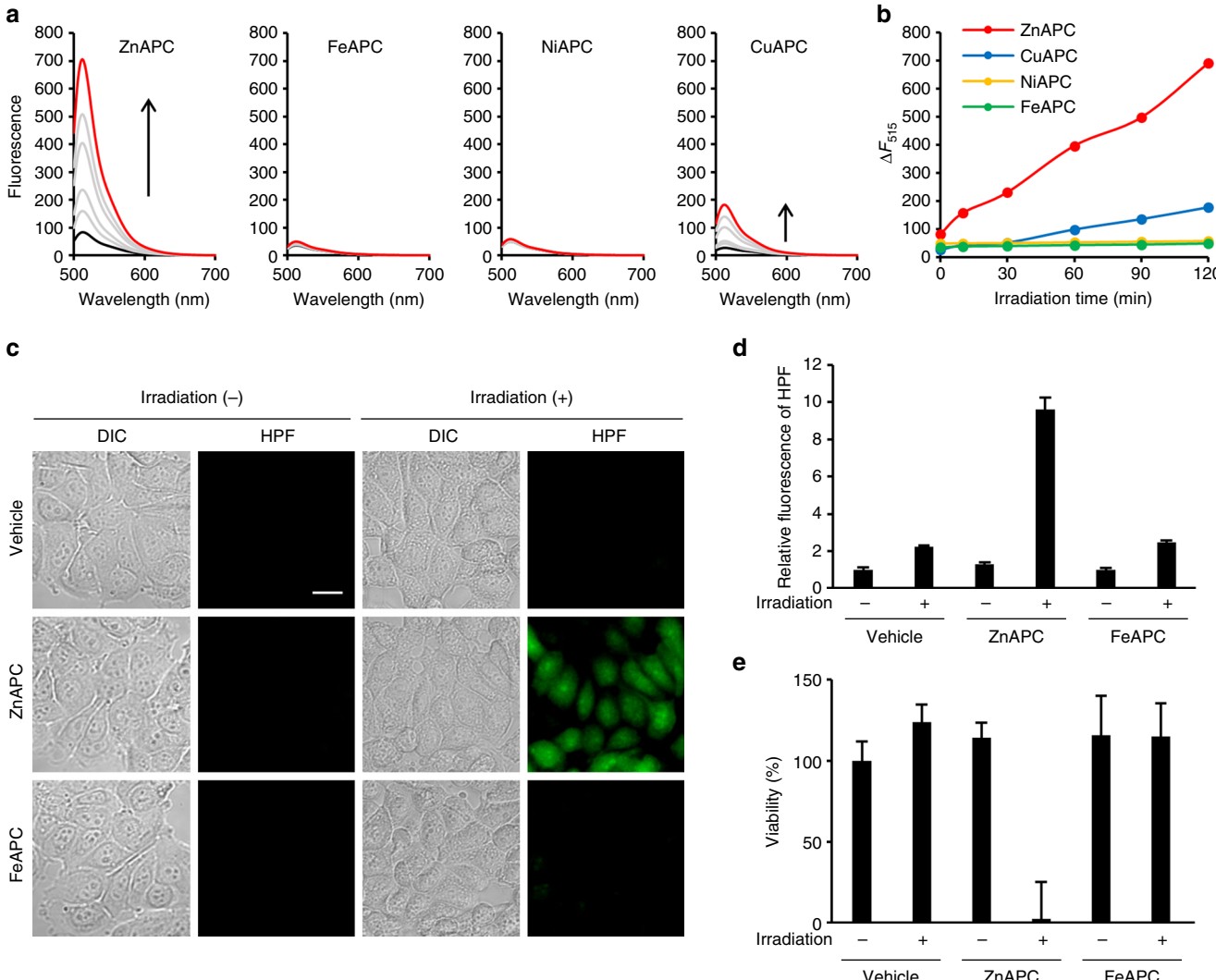

**Fig. 2** ZnAPC induces ROS production in vitro and in cells. **a** Fluorescence spectra of 10 μM HPF as an ROS indicator in the presence of 2 μM ZnAPC, FeAPC, NiAPC or CuAPC at 25 °C. Excitation wavelength was 490 nm. The spectra at irradiation time points 0 and 120 min are highlighted in black and red, respectively. **b** Plots of the change in fluorescence intensity of HPF at 515 nm during the photo-irradiation. **c** Evaluation of ROS levels by means of HPF in MCF-7 cells treated with ZnAPC or FeAPC after photo-irradiation for 1 h. DIC images of cells and fluorescent images of HPF are presented. Scale bar, 20 μm. **d** The mean fluorescence intensity of HPF was quantified. Relative fluorescence intensities are shown, and each bar represents mean ± SD for 10 images. **e** Cells pre-treated with ZnAPC or FeAPC for 1 h were incubated for 24 h after photo-irradiation for 2 h. The number of viable cells was determined by Trypan Blue exclusion–based cell staining. The number of treated cells was normalised to that of untreated cells. Each bar represents mean ± SD; $n = 3$. All intracellular experiments were conducted with 10 μM HPF, 10 μM ZnAPC or 10 μM FeAPC

drugs targeting DNA or RNA G-quadruplexes. This is at least partly because of non-specific binding of known ligands of G-quadruplexes to DNA and RNA duplexes, which are the most abundant structures of genomic DNA and coding and non-coding transcripts, respectively[36]. Moreover, almost all known ligands bind to a target G-quadruplex in a reversible manner, leading to only a transient effect of the ligand on the respective gene expression. The expression of cancer-related proteins regulated by formation of a G-quadruplex, including NRAS, can be effectively controlled if a ligand can not only bind reversibly but also attack irreversibly DNA and RNA G-quadruplexes.

Here, we demonstrate that an anionic phthalocyanines with $Zn^{2+}$ (ZnAPC) not only binds but also cleaves a G-quadruplex-forming 5′ UTR of the *NRAS* mRNA G-quadruplex after photo-irradiation and induces cell death. It is next shown that ZnAPC is a capable photosensitiser for direct transfer of energy to *NRAS*

mRNA and induces its breakdown upon photo-irradiation even under low-oxygen conditions, which are defining feature of solid tumours. The approach in this study holds promise for a molecularly targeted PDT for cancer.

## Results

**ZnAPC binds to the G-quadruplex derived from *NRAS* mRNA.** Some small molecules have been found to bind to a G-quadruplex in a selective manner[37–39]. Among these molecules, anionic phthalocyanines (APCs) coordinating $Ni^{2+}$ (NiAPC) or $Cu^{2+}$ (CuAPC) bind to DNA G-quadruplexes derived from human telomeric DNA[40,41]. According to these results, we hypothesised that anionic phthalocyanine derivatives can control NRAS expression by photo-irradiation. Because a phthalocyanine coordinating $Zn^{2+}$ has a high photosensitising ability among phthalocyanine derivatives[3], we used anionic phthalocyanines,

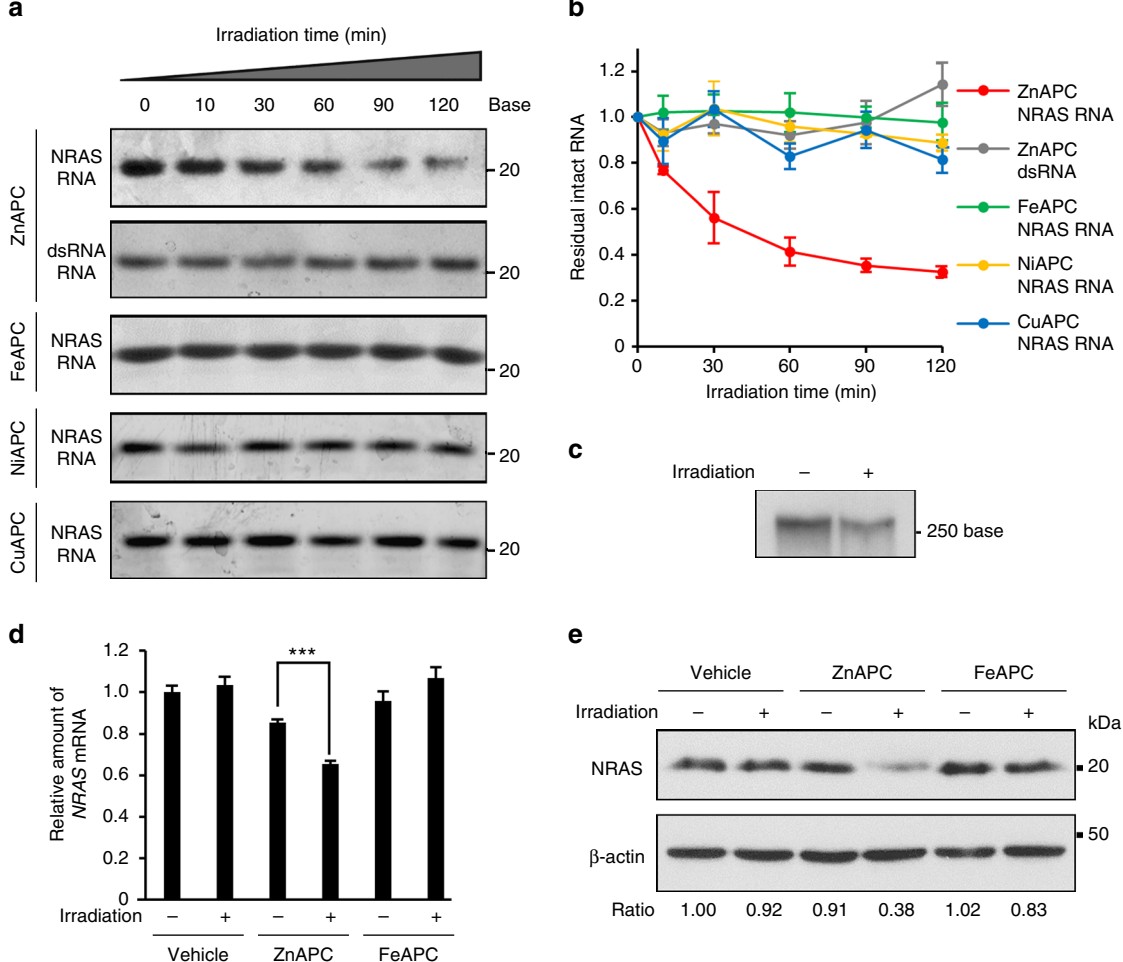

**Fig. 3** ZnAPC decreases the amount of *NRAS* mRNA. **a** Denaturing polyacrylamide gel electrophoresis (10% gel) of 0.1 μM NRAS RNA or dsRNA in the presence of 2 μM ZnAPC, FeAPC, NiAPC or CuAPC after photo-irradiation for the indicated periods at 25 °C. **b** Residual intact RNA after photo-irradiation of NRAS RNA and dsRNA in the presence of APCs. Error bars represent mean ± SD; n = 3. **c** 0.1 μM NRAS F-RNA labelled with DIG in the presence of 2 μM ZnAPC before and after photo-irradiation for 120 min were analysed by electrophoresis in a 15% denaturing polyacrylamide gel. The RNA was detected with antibody against DIG. **d**, **e** Cells pre-treated with 10 μM ZnAPC or FeAPC together with 1 μg ml⁻¹ actinomycin D, which inhibits de novo mRNA synthesis, for 1 h were photo-irradiated for 2 h. **d** The amount of *NRAS* mRNA was evaluated by real-time PCR. Each bar represents mean ± SD; n = 3. For statistical significance, an unpaired t-test was performed. ***p < 0.0001. **e** The cells were incubated for 5 h after photo-irradiation. The cell extracts were subjected to immunoblot analysis with antibodies against NRAS; β-actin served as a loading control. Blots of NRAS and β-actin were quantified, and the relative values of NRAS are shown

ZnAPC with zinc as a coordinated metal in addition to NiAPC, CuAPC and FeAPC with iron as a coordinated metal.

Because RNA is more abundant than its corresponding DNA in the cell, firstly, we evaluated the binding affinity of APCs for NRAS RNA, which is a parallel G-quadruplex-forming RNA oligonucleotide derived from the 5′ UTR of *NRAS* mRNA (the nucleotide sequence and circular dichroism (CD) spectrum are shown in Supplementary Table 1 and Supplementary Figure 1, respectively). Figure 1a shows the visible (VIS) absorption spectra of 2 μM APCs in the presence of various concentrations of NRAS RNA. In the case of ZnAPC, a broad absorption peak near 640 nm was observed in the absence of NRAS RNA. An increase in NRAS RNA concentration reduced the broad peak at 640 nm and concomitantly enhanced a new sharp peak at 680 nm. The broad peak at 640 nm and the sharp peak at 680 nm are absorption patterns of an oligomeric phthalocyanine and monomeric phthalocyanine, respectively, in an aqueous solution[36,40,42]. Hence, the spectral change of ZnAPC induced by the addition of NRAS RNA shows that the oligomer of ZnAPC dissociates into

the monomeric form by the binding to NRAS RNA. Figure 1b shows ΔAbsorbance (Abs. with RNA minus Abs. without RNA) of 2 μM APCs at 680 nm. The dissociation constant ($K_d$) of ZnAPC for binding to NRAS RNA was estimated to be 3.1 μM at 25 °C (see Materials and Methods for the procedure) (Fig. 1c). On the other hand, the absorbance spectrum of FeAPC was not altered by the addition of NRAS RNA (Fig. 1a, b), implying that FeAPC did not bind to NRAS RNA. Because the coordinated iron ion yields additional ligands at axial positions[43,44], FeAPC stacking is offset, which probably inhibits the π–π interaction between FeAPC and the G-quartet. Although further studies are necessary to elucidate why FeAPC does not bind to NRAS RNA, FeAPC may be useful as a control APC because of the low binding affinity for the target NRAS RNA. In the cases of both NiAPC and CuAPC, the higher NRAS RNA concentration enhanced a new sharp peak at 680 nm as shown in ZnAPC. In addition, a peak shift in the broad peak was observed, indicating that NRAS RNA is likely to promote the conversion from the oligomeric form into monomeric form of NiAPC and CuAPC via

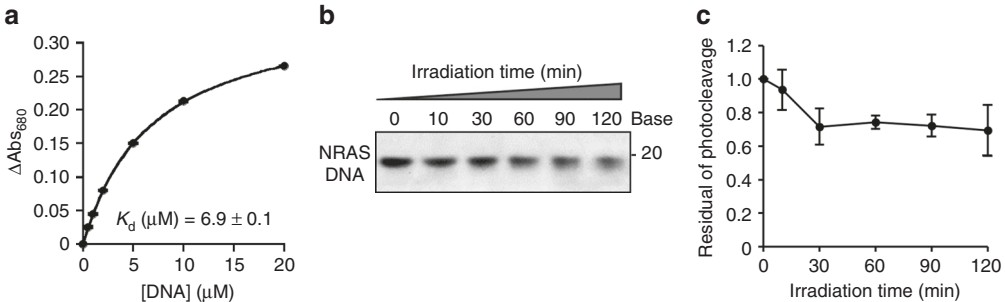

**Fig. 4** ZnAPC binds to NRAS DNA. **a** Plots of ΔAbsorbance at 680 nm ( = absorbance with NRAS DNA minus absorbance without NRAS DNA) of 2 μM ZnAPC with 0, 0.5, 1, 2, 5, 10 or 20 μM NRAS DNA at 25 °C. Continuous lines are curve-fitting results with a theoretical equation, and the estimated $K_d$ of ZnAPC with NRAS DNA is also indicated. **b** Electrophoresis of 0.1 μM NRAS DNA (in a 10% denaturing polyacrylamide gel) in the presence of 2 μM ZnAPC after photo-irradiation for the indicated periods. **c** Residual intact DNA after the photo-cleavage. Error bars represent mean ± SD; $n = 3$

an intermediate form. The $K_d$ values of NiAPC and CuAPC were estimated, with a two-state assumption as utilised for the $K_d$ evaluation of ZnAPC, to be 3.9 and 2.7 μM, respectively, at 25 °C (Fig. 1c), indicating that the binding affinity of NiAPC and CuAPC for NRAS RNA is similar to that of ZnAPC.

Considering that RNA duplexes is abundant in cells, the APCs should bind to NRAS RNA in the presence of an excess of double-stranded RNA (dsRNA). We also checked the binding of APCs to a dsRNA (the nucleotide sequence is shown in Supplementary Table 1). Almost no absorbance change was observed for all the APCs after addition of dsRNA, suggesting that the APCs did not bind to dsRNA under these experimental conditions. Given that CuAPC and NiAPC did not bind to DNA duplexes[40], the APCs is assumed not to bind to DNA and RNA duplexes. It was next shown that the binding of 2 μM ZnAPC to NRAS RNA was almost the same even in the presence of 100 μM dsRNA (ZnAPC spectrum, Fig. 1b). The value of $K_d$ was 4.3 μM at 25 °C, which is almost the same as the value in the absence of dsRNA. Taken together, these results support the possibility that ZnAPC, NiAPC and CuAPC, target the G-quadruplex derived from the 5′ UTR of *NRAS* mRNA in cells.

**ZnAPC generates ROS after photo-irradiation**. Phthalocyanines generate reactive oxygen species (ROS) after photo-irradiation, depending on the coordinated metal at the centre[45–47]. It is reasonable to hypothesise that the APCs used in this study generate ROS after photo-irradiation and effectively cleave the bound NRAS RNA. To determine whether the APCs can generate ROS in an aqueous solution, we utilised hydroxyphenyl fluorescein (HPF), which acts as a sensitive fluorescent probe for the detection of the most active ROS: the hydroxyl radical[48]. Fluorescence intensity of HPF at 515 nm increased after photo-irradiation of ZnAPC and CuAPC, although the increment of the fluorescence intensity yielded by ZnAPC was significantly higher than that of CuAPC (Fig. 2a, b). Contrary to ZnAPC and CuAPC, fluorescence intensity of HPF was almost unchanged for FeAPC and NiAPC. These results indicated that ZnAPC effectively generates ROS upon photo-irradiation compared with other APCs.

We next examined the effects of ZnAPC, as well as FeAPC as a negative control APC, on the ROS formation in human breast cancer MCF-7 cells, in which *NRAS* amplification has been reported[49]. Figure 2c shows fluorescent images of HPF in MCF-7 cells in the absence of APC and in the presence of ZnAPC or FeAPC with or without photo-irradiation. Fluorescence from HPF was observed only in the presence of ZnAPC after photo-irradiation, indicating that ROS formation by ZnAPC is significantly increased by the photo-irradiation. In addition,

fluorescence intensities of HPF in the presence of FeAPC with or without photo-irradiation were much weaker than those in the presence of ZnAPC. These fluorescent intensities, conclusively listed in Fig. 2d, show that the fluorescence intensity yielded by ZnAPC with photo-irradiation is significantly higher than that in the other cases. Moreover, measurement of cell viability by trypan blue exclusion-based cell staining revealed that ZnAPC markedly decreased cell viability after photo-irradiation (Fig. 2e and Supplementary Figure 2), whereas other conditions did not influence cell viability. These results suggested that ZnAPC generates ROS upon photo-irradiation and concomitantly induces cell death.

**ZnAPC photo-cleaves a G-quadruplex derived from *NRAS* mRNA**. From this point of view, we tested whether, upon photo-irradiation, ZnAPC degraded the NRAS RNA under the experimental conditions. Figure 3a presents denaturing polyacrylamide gel electrophoresis of NRAS RNA and dsRNA in the presence of APCs after various irradiation periods from 0 to 120 min. Only in the case of NRAS RNA in the presence of ZnAPC, upon photo-irradiation, did band intensity of the intact NRAS RNA decrease (Fig. 3a), and the cleaved NRAS RNA was detected (see Supplementary Figure 3a and b for the whole image of the gel and the cleaved product). On the other hand, reduction of the band intensity of dsRNA was not observed, in agreement with the properties of ZnAPC binding to dsRNA. Figure 3b shows relative residual amounts of RNAs after photo-irradiation-induced cleavage. The residual amount of NRAS RNA was significantly decreased during the photo-irradiation of ZnAPC to the value lower than 0.4, showing that more than 60% of NRAS RNA was cleaved after the photo-irradiation for 120 min. Furthermore, it was found that other anionic phthalocyanines, i.e. FeAPC, NiAPC and CuAPC, did not cleave NRAS RNA after photo-irradiation.

We next tested whether ZnAPC cleaved RNA oligonucleotides derived from the full sequence of the 5′ UTR of *NRAS* mRNA harbouring the G-quadruplex motif (NRAS F-RNA) (Supplementary Figure 4a and b). The synthesised DIG-labelled NRAS F-RNA by in vitro transcription was used after purification. As shown in the model sequence, NRAS RNA, the band intensity of NRAS F-RNA diminished after photo-irradiation (Fig. 3c). In addition, the products of photo-cleavage of NRAS F-RNA were clearly observed (Supplementary Figure 4c). These results indicate that NRAS F-RNA can be photo-cleaved by ZnAPC.

The results on the binding, on the ROS generation, and the photo-cleavage abilities indicate that ZnAPC is a promising photosensitiser targeting *NRAS* mRNA in living cells. Moreover,

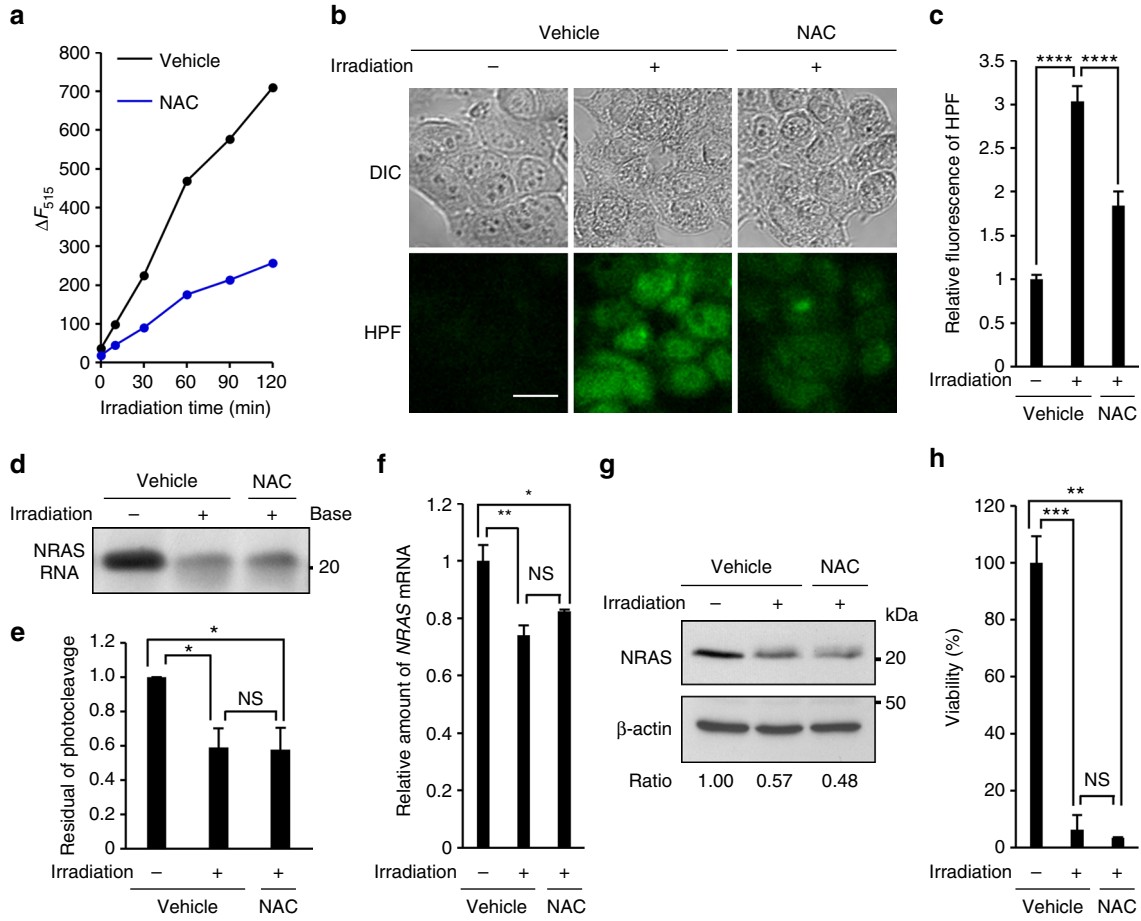

**Fig. 5** NAC does not inhibit downregulation of NRAS expression by ZnAPC. **a** Fluorescence spectra of 10 μM HPF in the presence of 10 mM NAC with 2 μM ZnAPC after photo-irradiation were acquired at 25 °C. Fluorescence intensities of HPF at 515 nm after photo-irradiation (0–120 min) are shown. **b**, **c** Cells pre-treated with 10 μM HPF together with 10 mM NAC for 30 min were treated with 10 μM ZnAPC for 30 min and subsequently photo-irradiated for 1 h. DIC images of cells and fluorescent images of HPF are presented in **b**. Scale bar, 20 μm. The mean fluorescence intensity of HPF was quantified. **c** Relative fluorescence intensities are shown, and each bar represents mean ± SD of 10 images. **d** Electrophoresis (in a 10% denaturing polyacrylamide gel) of 0.1 μM NRAS RNA in the presence of 2 μM ZnAPC with or without 10 mM NAC, followed by photo-irradiation for 2 h. **e** Residual intact RNA after the photo-cleavage of NRAS RNA. Each bar represents mean ± SD; n = 3. **f**, **g** Cells pre-treated with 10 μM ZnAPC together with 1 μg ml$^{-1}$ actinomycin D and 10 mM NAC for 1 h were photo-irradiated for 2 h. **f** The amount of *NRAS* mRNA was evaluated by real-time PCR. Each bar represents mean ± SD; n = 3. **g** The cells were incubated for 5 h after photo-irradiation. The cell extracts were subjected to immunoblot analysis with antibodies against NRAS; β-actin served as a loading control. Blots of NRAS and β-actin were quantified, and the relative values of NRAS are shown. **h** Cells pre-treated with 10 μM ZnAPC together with 10 mM NAC for 1 h were incubated for 24 h after photo-irradiation for 2 h. The number of viable cells was determined by Trypan Blue exclusion-based cell staining. The number of treated cells was normalised to that of untreated cells. Each bar represents mean ± SD; n = 3. For statistical significance, an unpaired *t*-test was performed. *$p < 0.01$; **$p < 0.001$; ***$p < 0.0001$; ****$p < 0.00001$; NS not significant, $p > 0.05$

we found that ZnAPC is capable of cell penetration and diffused throughout the cytosol in MCF-7 cells (Supplementary Figure 5). On the basis of ZnAPC, we next attempted to confirm that ZnAPC targets the G-quadruplex in the 5′ UTR of *NRAS* mRNA in living cells. Figure 3d shows a relative amount of *NRAS* mRNA in the absence and in the presence of ZnAPC or FeAPC with or without photo-irradiation. Treatment with ZnAPC did not influence the relative amount of *NRAS* mRNA without photo-irradiation. On the other hand, it was found that the 120 min photo-irradiation reduced it by ~35%. As expected, treatment with FeAPC had no influence on the amount of *NRAS* mRNA even after photo-irradiation. The specific reduction in the amount of *NRAS* mRNA by the combination of ZnAPC and photo-irradiation is in agreement with the photo-cleavage of NRAS RNA and NRAS F-RNA by ZnAPC as mentioned above. We next determined the expression level of the NRAS protein after the same treatments (Fig. 3e). The photo-irradiation in the absence of

ZnAPC did not reduce NRAS protein expression as shown in lane 2. In contrast, the expression level was significantly decreased by photo-irradiation in the presence of ZnAPC, up to the value of 0.38 (lane 4), but not in the presence of FeAPC (lane 6). These findings are consistent with the mRNA expression levels under the same conditions (Fig. 3d).

It has been reported that formation of a G-quadruplex in the 5′ UTR of *NRAS* mRNA impairs its translation[21]. Given that the expression of NRAS was not significantly downregulated by ZnAPC without photo-irradiation, ZnAPC is unlikely to promote stabilisation of the G-quadruplex in the 5′ UTR of *NRAS* mRNA in the cell. On the other hand, NRAS expression was decreased by ZnAPC after photo-irradiation, implying that ZnAPC bound and irreversibly photo-cleaved the G-quadruplex of *NRAS* mRNA in the cell.

Furthermore, we tested whether ZnAPC bound the G-quadruplexes of not only *NRAS* mRNA but also of corresponding

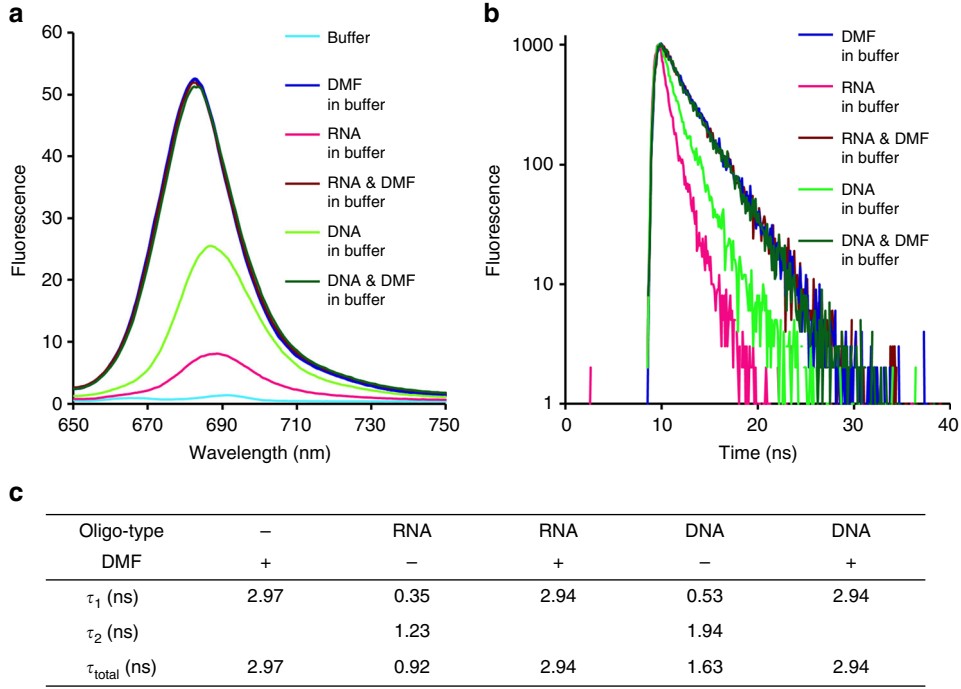

| Oligo-type | – | RNA | RNA | DNA | DNA |
|---|---|---|---|---|---|
| DMF | + | – | + | – | + |
| $\tau_1$ (ns) | 2.97 | 0.35 | 2.94 | 0.53 | 2.94 |
| $\tau_2$ (ns) | | 1.23 | | 1.94 | |
| $\tau_{total}$ (ns) | 2.97 | 0.92 | 2.94 | 1.63 | 2.94 |

**Fig. 6** Energy transfer from ZnAPC to NRAS RNA is greater than that to NRAS DNA. **a** Fluorescence spectra of 2 μM ZnAPC under the following experimental conditions: in the absence of NRAS RNA in the buffer (right blue) and in the buffer containing 30 wt% DMF (dark blue), in the presence of NRAS RNA in the buffer (red) and in a buffer containing 30 wt% DMF (brown), in the presence of NRAS DNA in the buffer (light green), and in the buffer containing 30 wt% DMF (dark green). **b** Fluorescence decay of ZnAPC under the experimental conditions listed in **a**. Fluorescence decay in the buffer is not shown because of the insufficient fluorescence intensity. All the experiments were conducted with 2 μM ZnAPC, 10 μM NRAS RNA, and 10 μM NRAS DNA. **c** Fluorescence lifetime, $\tau$, was evaluated with a theoretical equation of single or double exponential decay for the fluorescence decay obtained in **b**

DNA (NRAS DNA, the nucleotide sequence is shown in Supplementary Table 1). Figure 4a shows ΔAbs at 680 nm of 2 μM ZnAPC vs. NRAS DNA concentration in a titration experiment. It was found that the addition of NRAS DNA monomerised ZnAPC, and $K_d$ was evaluated and found to be 6.9 μM at 25 °C. The affinity of ZnAPC for NRAS DNA was of the same order of magnitude as that of NRAS RNA. It is noteworthy, however, that the degradation of NRAS DNA by ZnAPC was only ~30% after the 120 min photo-irradiation (Fig. 4b, c), whereas the degradation of NRAS RNA was ~70% as shown above (Fig. 3a, b). These results suggested that there is a specific mechanism of photo-cleavage of NRAS RNA.

**ZnAPC directly transfers electrons to NRAS RNA.** To elucidate the possible mechanism of cleavage of RNAS RNA by ZnAPC, we tested whether a ROS scavenger, *N*-acetyl cysteine (NAC), influences the photo-cleavage by NRAS RNA. Figure 5a shows a fluorescence intensity change of 10 μM HPF in the presence and absence of 10 μM NAC. It was confirmed that NAC decreased the fluorescence of HPF, corresponding to ROS production by ZnAPC with photo-irradiation. As shown in the test tube, NAC also reduced ROS production in the cell (Fig. 5b, c). NAC's effects on the photo-cleavage efficacy were further studied in the test tube and in the cell. In contrast to the ROS reduction, it was found by electrophoresis after the photo-irradiation that NAC did not affect photo-cleavage of NRAS RNA by ZnAPC (Fig. 5d, e). Furthermore, we demonstrated that NAC did not attenuate the downregulation of *NRAS* mRNA and NRAS protein expression by ZnAPC (Fig. 5f, g). Therefore, both in the test tube and in the cell, NAC reduces the ROS formation but does not inhibit the photo-cleavage. It was also found that NAC did not suppress

the cell death induced by ZnAPC after photo-irradiation (Fig. 5h). These results suggest that ZnAPC induces cell death via downregulation of NRAS regardless of ROS generation. Therefore, other factors, besides ROS production, are involved in the photo-cleavage of NRAS RNA and in the downregulation of NRAS by ZnAPC with photo-irradiation.

We therefore investigated by fluorescence lifetime measurement how ZnAPC cleaves the G-quadruplex of NRAS RNA. ZnAPC emits fluorescence when it is in the monomeric form[50,51]. In agreement with these previous reports, we found here that ZnAPC emitted a distinct fluorescent signal after the monomerisation and binding to NRAS RNA in the buffer, whereas fluorescence intensity was negligible without NRAS RNA (Fig. 6a). Fluorescence intensity of ZnAPC was also observed in a mixed solution (the buffer containing 30 wt% dimethylformamide [DMF]), in which ZnAPC is highly soluble and exists as a monomer (the absorption spectrum is given in Supplementary Figure 6). In the mixed solution, fluorescence intensities of ZnAPC in the presence of NRAS RNA or NRAS DNA were identical to that in the absence of the G-quadruplex. These fluorescence intensities were greater than those in the buffer, indicating that the mixed solution accelerated ZnAPC monomerisation. Figure 6b illustrates fluorescence decay of ZnAPC under these conditions. The fluorescence decay of ZnAPC without the G-quadruplex in the mixed solution could be fitted to a theoretical equation of a single exponential function, and fluorescence lifetime, $\tau_1$ ( $= \tau_{total}$), was estimated to be 2.97 ns. The $\tau_1$ values in the presence of NRAS RNA or NRAS DNA were the same as those without the G-quadruplex and consistent with the steady-state fluorescence spectra (Fig. 6a). It has been reported that fluorescence decay of Zn-phthalocyanine derivatives in DMF follows a single exponential function and that their

$\tau_{total}$ values are evaluated to be 1 to 4 ns, which correspond to the conversion from $S_1$ to $S_0$[52–54]. Thus, the $\tau_{total}$ values listed in Fig. 6c for the mixed solution are consistent with the other reports. Moreover, the fluorescence decay in the buffer in the presence of NRAS RNA could be fitted to a double exponential function but not to a single one. $\tau_1$ and $\tau_2$ were estimated to be 0.35 and 1.23 ns, respectively (Fig. 6c). The value of $\tau_{total}$ was 0.92 ns. Similarly, $\tau_1$ and $\tau_2$ were 0.53 and 1.94 ns for the double exponential function and the buffer with NRAS DNA. Because larger dipole moments of coexisting molecules can increase the energy transfer efficiency, fluorescence lifetime is generally shorter in a more polar environment. The value of $\tau_{total}$ in the presence of NRAS RNA was smaller than that with NRAS DNA, suggesting that smaller $K_d$ makes fluorescence lifetime shorter. Therefore, it is possible that $\pi$–$\pi$ stacking interactions with a large aromatic G-quartet led to a less polar environment of ZnAPC.

It has been reported that fluorescence decay of a porphyrin–DNA G-quadruplex complex has two lifetime components, whereas the porphyrin has a single lifetime component in its free state[55]. In that study, the authors concluded that two lifetime components indicate the existence of two binding states of the porphyrin because they also found that the binding stoichiometry of the porphyrin and DNA G-quadruplex is 2:1. In contrast to that porphyrin, it has been shown that the stoichiometry of CuAPC with a DNA G-quadruplex is 1:1[36]. Thus, a single binding mode of ZnAPC is not consistent with a possible mechanism involving two fluorescence lifetime components. Notably, it has been demonstrated that guanine mononucleotide and guanine-containing polynucleotides decrease fluorescence lifetime of a porphyrin[56–58]. This finding was interpreted as a manifestation of photo-induced electron transfer between the guanine base and the photoexcited porphyrin. In addition, the decrease in a fluorophore activity under the influence of photo-induced electron transfer was recently applied to monitoring a structure switch from a hairpin duplex to a G-quadruplex by measurement of the fluorescence reduction and thus shorter fluorescence lifetime[59]. According to those results, it is possible that the sub-nano-second $\tau_1$ values (0.35 ns for NRAS RNA and 0.53 ns for NRAS DNA) reflect the photo-induced electron transfer that directly proceeds between ZnAPC and a G-quadruplex.

It has been reported that H (hydrogen) abstraction from an RNA strand by the hydroxyl radical is much faster than the reaction with a hydroxyl radical scavenger, and consequently such a scavenger cannot inhibit the H abstraction from RNA[60]. It is therefore possible that $\tau_1$ of NRAS RNA, smaller than that of NRAS DNA (Fig. 6b, c), supports the direct energy transfer from ZnAPC to NRAS RNA rather than to NRAS DNA. Given that cleavage of NRAS RNA via photo-irradiation of ZnAPC is more effective than that of NRAS DNA (Figs. 3a, b and 4b, c), the direct energy transfer is likely to cause its photo-cleavage. Therefore, the longer and shorter fluorescence lifetimes of ZnAPC in the presence of the G-quadruplex can be attributed to the conversion from $S_1$ to $S_0$ and the photo-induced electron transfer, respectively. The photo-irradiation of NRAS RNA by photo-induced electron transfer via H abstraction from the RNA strand enables ZnAPC to selectively cleave the RNA G-quadruplex but not other biomolecules present in the vicinity of ZnAPC. The cleavage mechanism will be further discussed below along with photo-cleavage reactions under aerobic and anaerobic conditions.

## Discussion
Our findings presented in this study highlight the direct photo-cleavage of the G-quadruplex in *NRAS* mRNA by ZnAPC. On the other hand, it was also clearly shown that not all RNA G-quadruplexes are targeted by ZnAPC. To rationalise the selectivity of ZnAPC, we studied ZnAPC binding with various RNA G-quadruplexes. First, we evaluated the binding of ZnAPC to other naturally occurring G-quadruplex-forming RNA oligonucleotides (VEGF RNA and BCL2 RNA) derived from the 5′ UTR of mRNA encoding vascular endothelial growth factor (VEGF) and B-cell lymphoma 2 (Bcl2), respectively. Nucleotide sequences and CD spectra of these RNAs are given in Supplementary Table 1 and Supplementary Figure 1, respectively. Supplementary Figure 7a shows VIS absorption spectra of 2 μM ZnAPC in the presence of various concentrations of VEGF RNA or BCL2 RNA, respectively. The smaller increase in absorbance at 680 nm showed that VEGF RNA and BCL2 RNA produced lesser amounts of monomeric ZnAPC as compared with NRAS RNA (Supplementary Figure 7b). The values of $K_d$ were estimated to be 48 μM for VEGF RNA and 36 μM for BCL2 RNA at 25 °C. These results mean that the binding of ZnAPC to VEGF RNA or BCL2 RNA is weaker than the binding to NRAS RNA ($K_d$ = 3.1 μM at 25 °C, as shown in Fig. 1c). Similarly, fluorescence intensity changes of ZnAPC promoted by VEGF RNA and BCL2 RNA were smaller than the change driven by NRAS RNA (Supplementary Figure 7c). In addition to the titration experiments traced by a VIS absorbance change, $K_d$ values of ZnAPC with these RNAs were evaluated with more data points in a lower concentration range because sensitivity of fluorescence detection is much higher than that of absorbance change. Supplementary Figure 7d depicts the titration curves traced by the fluorescence and absorbance changes, revealing that the titration curves are almost identical to each other. The values of $K_d$ estimated based on these independent titration data were also very similar to each other (Supplementary Figure 7e). These results suggest that the monomerisation of ZnAPC and fluorescence enhancement, reflecting electron and energy transfers, correlate with each other. Furthermore, it was found that the photo-cleavage activity of ZnAPC towards VEGF RNA and BCL2 RNA is undetectable (Supplementary Figure 7f, g).

We wondered how the number of G-quartets and the nucleotide sequence in the loop regions affect the binding of ZnAPC to RNA G-quadruplexes. To address this point, we systematically designed several NRAS RNA mutants (NRAS MT1 RNA to NRAS MT5 RNA: nucleotide sequences are shown in Supplementary Table 1). Firstly, it was confirmed that all the RNAs formed a parallel G-quadruplex same as NRAS RNA (CD spectra are presented in Supplementary Figure 1). NRAS RNA has three G-quartet planes and relatively short loop regions, whereas VEGF RNA and BCL2 RNA have two G-quartet planes and longer loop regions, respectively. Based on these differences, NRAS MT1 RNA and NRAS MT2 RNA were designed to have the same loop sequences as do VEGF RNA and BCL2 RNA, respectively. The $K_d$ values of ZnAPC for NRAS MT1 RNA and NRAS MT2 RNA, evaluated by the VIS absorbance changes, were estimated to be 9.2 and 4.2 μM, respectively, at 25 °C (Supplementary Figure 8). These values are comparable to that of NRAS RNA (3.1 μM, as shown in Fig. 1c), implying that high affinity of ZnAPC for NRAS RNA is not dependent on the loop sequences. NRAS MT3 RNA maintains the loop sequence but consists of only two G-quartet planes. $K_d$ was estimated to be 48 μM at 25 °C (Supplementary Figure 8c), which is more than 10-fold greater than that of NRAS RNA. This finding shows that ZnAPC preferentially binds the G-quadruplex consisting of three rather than two G-quartets. Although the G-quadruplex structure of BCL2 RNA consists of three G-quartet planes, the binding affinity for ZnAPC is low. Because the number of guanines at the 5′ and 3′ ends of BCL2 RNA was greater than that of NRAS RNA, we next examined the binding for NRAS MT4 RNA and NRAS MT5 RNA, which have one more guanine nucleotide and uridine

nucleotide at both ends of NRAS RNA, respectively. The $K_d$ values of ZnAPC with NRAS MT4 RNA and NRAS MT5 RNA at 25 °C were estimated to be 95 and 4 μM, respectively, revealing that the additional guanine nucleotides but not uridine nucleotides significantly decreased the affinity for ZnAPC. Considering that the sequence of NRAS MT5 RNA is derived from *NRAS* mRNA (refer to the sequence depicted in Supplementary Figure 4b), not only three G-quartets but also the nucleotides next to the G-quadruplex–forming region may be important for strong affinity for ZnAPC. Although further studies are required, these results suggest that the number of G-quartet planes and the flanking sequences at the 5′ and 3′ ends contribute to the sequence-selective binding and photo-cleavage of target NRAS RNA and *NRAS* mRNA by ZnAPC.

It is known that hypoxia, a common characteristic of malignant tumours, promotes invasion and metastasis of cancer cells and often causes resistance to chemotherapy and radiotherapy, and this resistance is mediated by hypoxia-inducible factors (HIFs)[61,62]. Given that RAS induces HIF-1α expression, small molecules targeting RAS during hypoxia will be effective as therapeutic agents against cancer. Here, we found that ZnAPC photo-cleaves the G-quadruplex in *NRAS* mRNA even in the presence of NAC, an ROS scavenger (Fig. 5d–g). It was therefore expected that the photo-cleavage activity of ZnAPC for the RNA G-quadruplex can be attained even under anaerobic conditions. To test this notion, we compared the photo-cleavage efficiency under aerobic and anaerobic conditions. It was demonstrated that the photo-cleavage by ZnAPC under the anaerobic conditions was comparable with that under the aerobic conditions (Supplementary Figure 9). Because oxygen molecules are required for ROS generation, this photo-cleavage in the anaerobic condition suggests that the photo-induced direct energy transfer from ZnAPC to NRAS RNA plays a critical role in the photo-cleavage reaction. Taken together, these results indicate that ZnAPC is expected to suppress aggressive phenotypes of cancer cells via downregulation of RAS signalling pathways even when cancer cells not only are resistant to ROS but also are under hypoxic conditions.

In conclusion, we show here that NRAS expression can be downregulated through targeting the G-quadruplex in *NRAS* mRNA by ZnAPC. Studies using both melanoma cells and melanoma tumour-bearing mice by the group of Luigi Xodo have successfully shown that porphyrin C14H28-alkyl derivative C14, a photosensitiser identified as a G-quadruplex binding compound, induces growth arrest of cancer cells through downregulation of *KRAS* by targeting the G-quadruplex in its mRNA[10,11]. In these articles, it was shown that C14 inhibits translation of *KRAS* mRNA even without photo-irradiation and induces light-mediated cleavage of *KRAS* mRNA, which is thought to be associated with its ability to generate ROS. By combining these results on KRAS and our results on NRAS in this study, we can propose that the specific binding and photo-cleavage of G-quadruplex structures of *RAS* mRNAs by RNA G-quadruplex ligands are promising as photosensitisers in a molecularly targeted PDT for downregulation of a RAS signalling pathway.

## Methods

**Materials and antibodies**. All the high-pressure liquid chromatography (HPLC)-grade DNA and RNA oligonucleotides analysed in this study were acquired from Hokkaido System Science Co., Ltd. and Sigma-Aldrich Japan and used without further purification. Single strand concentrations of the oligonucleotides were determined by measuring the absorbance at 260 nm and high temperature using a UV-1800 spectrometer (Shimadzu, Kyoto, Japan). Single-strand extinction coefficients were calculated from mononucleotide and dinucleotide data using the nearest-neighbor approximation[63]. ZnAPC [zinc(II) phthalocyanine 3,4′,4″,4‴-tetrasulfonic acid, tetrasodium salt] was purchased from Frontier Scientific,

FeAPC [iron(III) phthalocyanine-4,4′,4″,4‴-tetrasulfonic acid, monosodium salt, compound with oxygen, hydrate], NiAPC [nickel(II) phthalocyanine-tetrasulfonic acid tetrasodium salt] and CuAPC [copper(II) phthalocyanine-3,4′,4″,4‴-tetra-sulfonic acid, tetrasodium salt] were purchased from Sigma-Aldrich. Actinomycin D and NAC were bought from WAKO Pure Chemical Industries, Ltd. Hydro-xyphenyl fluorescein (HPF) was purchased from Goryo Kayaku Co., Ltd. Anti-N-Ras mouse monoclonal (Santa Cruz Biotechnology, sc-31; 1:100) and anti-β-actin (Sigma-Aldrich, A1978; 1:1000) mouse monoclonal antibodies served as a primary antibody. A horseradish peroxidase (HRP)-conjugated sheep anti-mouse IgG antibody (Amersham Pharmacia, NXA931V; 1:10,000) served as a secondary antibody.

**Absorption spectroscopy**. Visible (Vis) absorption spectra for ZnAPC (2 μM) and FeAPC (2 μM) were recorded on a spectrophotometer (UV-1800; Shimadzu, Kyoto, Japan) with a quartz cell with 1 cm path length. All the measurements were carried out in a buffer consisting of 50 mM 2-(N-morpholino) ethanesulfonate [MES]-LiOH (pH 7) and 100 mM KCl at 25 °C. Before measurement, each sample was heated to 90 °C for 5 min and gently cooled to 25 °C at 0.5 °C min⁻¹.

**Fluorescence spectroscopy**. Fluorescence spectra for 2 μM ZnAPC in the presence of 0–20 μM RNA in a buffer consisting of 50 mM MES-LiOH (pH 7) and 100 mM KCl (ex: 620 nm, em: 650–750 nm) were recorded on a FP-8200 spec-trofluorometer (JASCO, Tokyo, Japan) equipped a temperature controller with a 0.3 cm × 0.3 cm quartz cell at 25 °C. Before measurement, each sample was heated to 90 °C for 5 min and gently cooled to 25 °C at 0.5 °C min⁻¹.

**Photo-irradiation**. For photo-irradiation of phthalocyanines in vitro, ZnAPC, FeAPC, NiAPC or CuAPC in a buffer consisting of 50 mM MES-LiOH (pH 7) and 100 mM KCl was irradiated with a light-emitting diode (LED) light with peak emission wavelength of 615 nm (PFBR-150RD-MN; 141.1 J cm⁻², CCS Inc., Kyoto, Japan). For photo-irradiation of phthalocyanines in the cell, cells were seeded at $5 \times 10^4$ cells cm⁻² and cultured overnight. The medium was then replaced with a CO₂-independent medium (Gibco) supplemented with 10% of foetal bovine serum and 1% of the penicillin/streptomycin solution. The cells were pre-treated with ZnAPC or FeAPC for 30 min or 1 h at 37 °C and next irradiated with LED light with peak emission wavelength of 630 nm (TH-160 × 120RD; 5.5 J cm⁻², CCS Inc., Kyoto, Japan).

**Evaluation of ROS formation**. An in vitro assay of ROS formation after the photo-irradiation of the APCs, fluorescence for 10 μM HPF in the presence of 2 μM ZnAPC, FeAPC, NiAPC or CuAPC in a buffer consisting of 50 mM MES-LiOH (pH 7) and 100 mM KCl was recorded at 25 °C (ex: 490 nm, em: 500–700 nm); an FP-8200 spectrofluorometer (JASCO, Tokyo, Japan) connected to a temperature controller was used with a a quartz cell with 0.3 cm path length. Before mea-surement, each sample was heated to 90 °C for 5 min and gently cooled to 25 °C at 0.5 °C min⁻¹. To evaluate the ROS formation in the cells, cells cultured on 35 mm glass bottom dishes (Matsunami Glass Ind., Ltd.) were pre-treated with ZnAPC or FeAPC for 30 min, followed by 10 μM HPF for 30 min. After photo-irradiation for 60 min, the cells were washed with PBS. A 488 nm laser was employed for imaging. Images were acquired by means of a confocal microscope (LSM700; Zeiss, Tokyo, Japan). The mean fluorescence intensity of HPF was quantified in the ImageJ software.

**Cleavage of RNA and DNA oligonucleotides**. The mixtures of 2 μM ZnAPC or FeAPC and 0.1 μM RNA or DNA in a buffer consisting of 50 mM MES-LiOH (pH 7) and 100 mM KCl were heated to 90 °C for 5 min and gently cooled to 25 °C at 0.5 °C min⁻¹. After the photo-irradiation of the mixtures in a quartz cell with 0.1 cm path length, a 1.5-fold volume of a stop solution (80 wt% formamide, 10 mM Na₂EDTA, and 0.01% blue dextran) was added. The samples containing 0.6 pmol RNA or DNA were then analysed by electrophoresis in a 10 or 15% polyacrylamide gel with 7 M urea at 70 °C. After that, the gels were stained with SYBR® Gold (Invitrogen), and the signals were quantified with a fluorescent imager, Typhoon FLA-9500 (GE Healthcare, Tokyo, Japan). When the efficiency of RNA cleavage was compared with that under aerobic and anaerobic conditions, the mixture of 2 μM ZnAPC and 0.1 μM NRAS RNA was dried before we dissolved it in the buffer.

To conduct the reaction under anaerobic conditions, the buffer was purged with argon for 5 min and then frozen by means of liquid N₂ followed by decompressing and purging with Ar gas three times. After thawing, the mixtures were dissolved in the buffer in the argon-equilibrated glove box. Uncropped scans of gels are shown in Supplementary Figure 10.

**Cleavage of NRAS F-RNA**. The pUC57 NRAS F-RNA vector containing the sequence encoding the 5′ UTR of *NRAS* mRNA following T7-promoter was constructed (GenScript Japan Inc.). The vector was digested with EcoRV, and the resultant DNA fragments were next separated by electrophoresis in a 1% agarose gel. The fragment containing DNA encoding the 5′ UTR of *NRAS* mRNA was purified by ethanol precipitation after it was extracted from the agarose gel with the GenElute™ Gel Extraction Kit (Sigma-Aldrich). This DNA served as a template

for in vitro RNA transcription. RNA was labelled with DIG and detected using the DIG RNA Labelling Kit (Roche Biochemicals). Briefly, the transcription reaction driven by T7 RNA polymerase in a reaction mixture containing DIG-11-UTG and an RNase inhibitor was carried out at 42 °C for 1 h. After treatment with DNase I at 37 °C for 15 min, RNA was purified by ethanol precipitation after phenol/chloroform extraction. A 1.5-fold volume of a stop solution was added to the RNA solution, and the mixture was subjected to electrophoresis in a 5% polyacrylamide gel with 7 M urea at 70 °C. The band of ~260 bases corresponding to NRAS F-RNA was cut out, and the RNA was purified by ethanol precipitation after extraction in TE buffer by rotation at 4 °C for 2 h. The mixtures of 2 µM ZnAPC and 0.1 µM RNA in a buffer consisting of 50 mM MES-LiOH (pH 7) and 100 mM KCl were heated to 90 °C for 5 min and gently cooled to 25 °C at 0.5 °C min$^{-1}$. After photoirradiation of the mixtures in a quartz cell with 0.1 cm path length, a 1.5-fold volume of a stop solution was added. The samples containing 0.6 pmol RNA were then analysed by electrophoresis in a 15% polyacrylamide gel with 7 M urea at 70 °C. After that, RNAs in the gel were transferred for 1 h at 100 mA in a cold room to a Hybond N$^{+}$ membrane (Amersham Biosciences) using 0.5 × TBE (Tris-borate-EDTA) and fixed by UV irradiation for 3 min. The membrane was washed with washing buffer and then blocked with blocking buffer for 30 min. After incubation with anti-DIG-AP (Roche Biochemicals, 12039672910; 1:10,000) for 30 min at room temperature, the membrane was washed with washing buffer and incubated with detection buffer for 30 min. The bound antibodies were visualised with a chemiluminescent substrate, CDP-Star (Roche Biochemicals) and subsequently detected by exposing blots to X-ray film (Fujifilm).

**Evaluation of the dissociation constant**. Absorption spectra for 2 µM ZnAPC or 2 µM FeAPC at various concentrations of RNA or DNA in a buffer consisting of 50 mM MES-LiOH (pH 7) and 100 mM KCl were recorded from 550 to 750 nm. A Shimadzu UV-1800 spectrophotometer connected to a Shimadzu TMSPC-8 thermoprogrammer (Shimadzu, Kyoto, Japan) was used with a quartz cell with 1 cm path length. Before measurement, each sample was heated to 90 °C for 5 min and gently cooled to 25 °C at 0.5 °C min$^{-1}$.

The dissociation constants ($K_d$) of ZnAPC with RNA or DNA oligonucleotides were evaluated by a curve-fitting procedure for the plot of ΔAbs [=(absorbance at 680 nm in the presence of the oligonucleotide) minus (absorbance at 680 nm in the absence of the oligonucleotide)] versus concentration of the oligonucleotide by means of the following equation at 25 °C:

$$\Delta Abs = \Delta Abs_{max} \times [\text{oligonucleotide}]/(K_d + [\text{oligonucleotide}]) \qquad (1)$$

where $\Delta Abs_{max}$ is the maximum difference in ΔAbs, and [oligonucleotide] is concentration of the added oligonucleotide. Note that the values of $\Delta Abs_{max}$ for BCL2 RNA and VEGF RNA were adjusted to the same value as that for NRAS RNA to evaluate $K_d$ values of ZnAPC because the $\Delta Abs_{max}$ values for these RNAs were too small to fit the data under our experimental conditions.

**Evaluation of fluorescence lifetime**. The fluorescencedecay curves of 2 µM ZnAPC or FeAPC in the presence or absence of 10 µM NRAS RNA or NRAS DNA was recorded on a fluorescence lifetime spectrometer (Quantaurus-Tau, C11567, Hamamatsu Photonics, Shizuoka, Japan) equipped with a photon counting unit (TDC unit, M12977-01, Hamamatsu Photonics). The decay curve measurements were carried out in a buffer consisting of 50 mM MES-LiOH (pH 7) and 100 mM KCl with or without 30 wt% DMF at room temperature. The fluorescence lifetime was evaluated by a curve-fitting system (U11487-01, Hamamatsu Photonics).

**Cell culture**. MCF-7 human breast cancer cells provided by the American Type Culture Collection (ATCC) were cultured in Dulbecco's modified Eagle's medium (Nissui Pharmaceutical) supplemented with 10% of foetal bovine serum (Sigma-Aldrich) and 1% of a penicillin/streptomycin solution (Wako Pure Chemical Industries).

**Immunoblot analysis**. Cells cultured on 60 mm dishes were disrupted with lysis buffer (50 mM Tris-HCl pH 7.4, 150 mM NaCl, 1% Triton X-100, 1% SDS, 10 mM EDTA, 1 mM Na$_3$VO$_4$, 10 mM NaF, and a protease inhibitor cocktail [PIC; Nacalai Tesque]) and then centrifuged at 20,000×$g$ for 10 min after sonication. The supernatants served as total cell extracts and were subjected to sodium dodecyl sulphate polyacrylamide gel electrophoresis (SDS-PAGE). Proteins were transferred to PVDF membranes and blocked for 30 min with 5% low-fat milk (Sigma-Aldrich) in TBS-T before the addition of primary antibodies. After incubation overnight at 4 °C or one hour at room temperature, the membrane was washed with TBS-T and incubated with an HRP-conjugated secondary antibody for 30 min. Bound antibodies were visualised with the HRP Chemiluminescent Reagent (Perkin Elmer) and subsequently detected by exposing blots to X-ray film (Fujifilm). Uncropped scans of immunoblots are shown in Supplementary Figure 10.

**Real-time PCR**. Cells were cultured on 60 mm dishes. Total RNA was extracted with the NucleoSpin RNA Plus Kit (Takara Bio Inc.), and cDNA was prepared with the PrimeScript 1st strand cDNA Synthesis Kit (Takara Bio Inc.). Real-time PCR analysis was performed on a StepOnePlus Real-Time PCR System (Applied

Biosystems, Foster City, CA, USA). Conditions for each transcript were as follows: 1 min at 95 °C, then 40 cycles at 95 °C for 15 s and 55 °C for 1 min. The following primers were used: human *NRAS* forward 5′-CAGAGGCAGTGGAGC TTGA -3′ and reverse 5′-GCTTTTCCCAACACCACCT-3′[64] and human *B2M* forward 5′-GCATTCCTGAAGCTGACA-3′ and reverse 5′-CGTGAGTAAACCT GAATCTTT-3′.

**Circular dichroism spectra**. CD spectra of oligonucleotides were measured for 20 uM oligonucleotide in the buffer consisting of 50 mM MES-LiOH (pH 7) and 100 mM KCl using a Jasco J-820 specroplarimeter (JASCO, Tokyo, Japan) with a quartz cell with 0.1 cm path length. Before measurement, each sample was heated to 90 °C for 5 min and gently cooled to 25 °C at 0.5 °C min$^{-1}$.

**Fluorescence microscopy**. Cells were cultured on a 35 mm glass bottom dish (Matsunami Glass Ind., Ltd.). ZnAPC incorporated into the cells was visualised because of its autofluorescence (ex: 488 nm and em: >630 nm). Images were acquired using a confocal microscope (LSM700; Zeiss) and then analysed in the ImageJ software (NIH).

**Statistical analysis**. Statistical analysis of data was performed by unpaired Student's two-sided *t*-test.

**Data availability**. The cell lines and all data supporting the findings of this study are available from the authors upon reasonable request.

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

## Acknowledgements

We thank Dr. Tamaki Endoh for discussion, and Ms. Sachi Wakasugi for carrying out purification of DIG-labelled RNA. This work was supported by JSPS KAKENHI (Grant Numbers 16K14042 and 15H03840 for K.K. and D.M.), especially a Grant-in-Aid for Scientific Research on Innovative Areas "Chemistry for Multimolecular Crowding Bio-systems" (17H06351 for D.M. and N.S.), the Naito Foundation Natural Science Scholarship for D.M. and the Asahi Glass Foundation Research Grant for D.M.

## Author contributions

K.K., H.T.-K., N.S. and D.M. designed the study. K.K. and D.M. with contributions from W.S., T.Y., K.M., K.I., K.T., T.T., K.A. and H.T.-K. performed all experiments. K.K. and D.M. wrote the manuscript.

## Additional information

**Competing interests:** The authors declare no competing interests.

