## [Peer Review File · Nature Communications]

Reviewers' comments:

Reviewer #1 (Remarks to the Author):

The manuscript by Kawauchi et al. describes the use of a phthalocyanine derivative to control NRAS mRNA level by targeting a G-quadruplex (G4) motif present within its 5'-UTR and inducing the degradation of the transcript by photo-irradiation. This is an interesting approach since RAS overexpression is associated with aggressive cancer phenotypes and that it has been proven difficult to target the RAS proteins. Nevertheless such approach has already been proposed by the group of Luigi Xodo and its feasibility demonstrated in a series of publication (see Faudale et al. *Chem Commun*, 2012, 48, 874-876; Rapozzi et al. *Molecular Cancer*, 2014, 13:75). Surprisingly these articles are neither discussed nor cited in the current article. It is difficult to identify a major conceptual advance and I suggest the paper may be a better fit for a more specialized journal.

Here are some questions/comments concerning the manuscript:

(i) It is not clear why the authors focused only on the Zn and Fe complexes of the phthalocyanine-tetrasulfonic acid compound. Studying the impact of other metal (Cu, Ni..) on the binding and photolysis of G4s will be of interest. The statement "The coordinated iron ion has additional ligands on both sides of FeAPC's plane" is incorrect. Prior oxidation, Fe(II)APC is planar and the absence of binding can't be explained by the presence of additional ligands.

(ii) Binding affinity to G4 motifs have been indirectly assessed using UV-spectroscopy quantifying the ratio between the monomeric/aggregated forms of the phthalocyanine derivatives. The reported dissociation constants are therefore describing the effect of the G4 on this equilibrium and do not solely reflect binding. Fluorescence spectroscopy, using low concentration of the G4s, could be used to this end. This point is crucial in order to understand why ZnAPC is able to degrade NRAS but not BCL2 and VEGF G4s.

(iii) It is difficult to understand why the authors extensively discuss the ability of ZnAPC to induce ROS since they clearly demonstrate that ROS formation does not contribute to downregulation of NRAS expression. Nonetheless Figure 2 reports experiments to support that "ROS production by photo-irradiation of ZnAPC is accelerated by its binding to the RNA G-quadruplex in MCF-7 cells". The reported data are not supporting this conclusion. Control experiments in which the level of the NRAS mRNA is modulated and showing different regime of ROS production are needed to support this conclusion.

(iv) Figure 3a-b reports the ability of ZnAPC to degrade the NRAS G4. It will be interesting to assess how "clean" is the photolysis reaction by studying longer sequences (full UTR) embedding the G4 motifs.

(v) The authors assess the binding and photolysis properties of ZnAPC on the NRAS DNA G4. The molecule, being polyanionic, is probably mainly localised in the cytoplasm of the cells. Did the authors assess where ZnAPC accumulate in MCF7 cells?

(vi) While the authors clearly ruled out the role of ROS formation on the downregulation of NRAS expression, the evidences for a direct electron transfer to the G4 motif remain scarce. To demonstrate C2'-hydrogen abstraction on the RNA substrate, a plausible hypothesis to explain the absence of activity on the DNA substrate, one could study the degradation products of the photolysis reaction under aerobic and anaerobic conditions (see Paul et al. *JACS* 2015, 137, 596-599 for example).

Reviewer #2 (Remarks to the Author):

Dear Authors,

Manuscript ID: 154934 entitled "An anionic phthalocyanine decreases NRAS expression by breaking down its RNA G-quadruplex" Keiko Kawauchi, Takatoshi Yasui, Kohei Murata, Wataru Sugimoto, Katsuhiko Itoh, Kazuki Takagi, Takaaki Tsuruoka, Hisae Tateishi-Karimata, Naoki Sugimoto, Daisuke Miyoshi is within the scope of Nature Communications. In this study, It was shown that expression of NRAS, signalling pathways contributes to aggressive phenotypes of cancer cells, could be controlled by photo-irradiation with an anionic phthalocyanine derivative with in vitro experiments and photophysical measurements. PDT is a relatively new cytotoxic treatment, predominantly used in anti cancer approaches. One of the most important issues in photodynamic cancer therapy is selectively accumulation of the photosensitizers in only cancer cells. Third-generation photosensitizers which is selective to cancer cell have been developed with different approaches based on targeting strategy. The approach in this study can be accepted as novel and promising for a molecularly targeted photodynamic cancer therapy. Therefore, the manuscript is acceptable for this journal.

Reviewer #3 (Remarks to the Author):

An anionic phthalocyanine decreases NRAS expression by breaking down its RNA G-quadruplex

The manuscript by Kawauchi et al describe the effect of a G-quadruplex binder, ZnAPC, on NRAS mRNA expression upon photo-irradiation.

The authors have previously shown that phthalocyanines are specific G-quadruplex ligands. Phthalocyanines have also been shown by others to generate reactive oxygen species (ROS) upon photo-irradiation. In this manuscript the authors hypothesized that photo-irradiation of ZnAPC and generation of ROS would induce cleavage and degradation of specific mRNAs. The hypothesis is interesting.

In the manuscript, the authors demonstrate that in vitro ZnAPC binds a G-quadruplex forming RNA sequence derived for the oncogenic NRAS mRNA but not G-quadruplex sequences derived from other mRNAs such as BCL2 and VEGF or double-stranded RNA sequences, suggesting that the binding is selective.

They further show that addition of ZnAPC followed by photo-irradiation reduces the amount of NRAS mRNA but not of BCL2 and VEGF mRNAs in MCF7 cells and induces cell death.

Overall, the results are clearly presented and convincing.

However, some aspects should be clarified

Major points:

1- The introduction is very short and inadequately referenced (mainly review articles). Of relevance, the role of NRAS in cancer is not described, the role of RNA G-quadruplexes in mRNA translation is not mentioned, and more importantly the literature describing the presence and role of G-quadruplexes in the NRAS 5'UTR (for example Kumari et al, Nat Chem Biol, 2007) is not mentioned.

Similarly, the manuscript should contain a discussion that describe the results in the context of the literature.

2- The results in Figure 2 are slightly striking. Figure 2C shows a very modest increase in ROS production in the presence of NRAS RNA in vitro. However, the results in cells presented in Figure

2D show a very strong increase in ROS production following photo-irradiation. This challenges the selectivity of ZnAPC for NRAS RNA as claimed by the authors.

3- Overall, and in many places, the authors claim that ZnAPC is selective for NRAS G-quadruplex sequence. However, this claim is only supported by the fact that ZnAPC does not bind two other G-quadruplex forming sequences (BCL-2 and VEGF) or one sequence of double-stranded RNA. There are thousands of putative G-quadruplex forming sequences in the 5'UTR of mRNAs. The data presented are not convincing to claim that ZnAPC is selective for ZnAPC.

4- The author claim that ZnAPC combined with photo-irradiation could have potential for cancer treatment. It would be useful if the authors could provide more information on the toxicity profile of ZnAPC. They observed that ZnAPC combined with photo-irradiation leads to cancer cell death, but how selective for cancer cell is this?

Minor points:

1- The gel portions presented in Figure 3 should be presented in full in supplementary material. If the photocleavage is specific, the decrease in NRAS RNA observed in Figure 3A should be balanced by the presence of bands of shorter length corresponding to the cleaved products.

Responses to the reviewers' comments

Ms No.: NCOMMS-18-01146A

Title: An anionic phthalocyanine decreases NRAS expression by breaking down its RNA G-quadruplex

Authors: Keiko Kawauchi, Wataru Sugimoto, Takatoshi Yasui, Kohei Murata, Katsuhiko Itoh, Kazuki Takagi, Takaaki Tsuruoka, Hisae Tateishi-Karimata, Kensuke Akamatsu, Naoki Sugimoto, Daisuke Miyoshi

We are grateful for the invaluable comments and suggestions made by the referees. In accordance with their comments, we performed additional experiments to address the issues raised and have made amendments to the manuscript. Please find our point-by-point responses to each of their comments. We appreciate the helpful suggestions from the referees, as their comments have strengthened our claim and significantly improved this paper.

Reviewer #1

Point 1: Nevertheless such approach has already been proposed by the group of Luigi Xodo and its feasibility demonstrated in a series of publication (see Faudale et al. Chem Commun, 2012, 48, 874-876; Rapozzi et al. Molecular Cancer, 2014, 13:75). Surprisingly these articles are neither discussed nor cited in the current article.

Response 1

We sincerely apologize for this issue. We now mention and cite these two important publications in the sections "Introduction" and "Discussion" in the revised manuscript (page 3, lines 12-13, and page 23, lines 7-16). By combining with their excellent results, here it is possible to propose that the specific binding and photo-cleavage of G-quadruplex structures of RAS mRNAs by NA G-quadruplex ligands are promising as photosensitizers in a molecularly targeted PDT for downregulation of a RAS signaling pathway.

Point 2: It is not clear why the authors focused only on the Zn and Fe complexes of the phthalocyanine-tetrasulfonic acid compound. Studying the impact of other metal (Cu, Ni..) on the binding and photolysis of G4s will be of interest.

Response 2

We agree with this comment. According to the comment, we examined the impact of NiAPC and CuAPC on the binding and photolysis of NRAS RNA, as well as ZnAPC and FeAPC. Although both NiAPC and CuAPC bound to NRAS RNA, it was found that they did not induce the cleavage of NRAS RNA upon photo-irradiation. We also showed that the efficiencies of ROS generation by NiAPC and CuAPC were significantly lower than the rate of ZnAPC. We have presented these results in the revised manuscript (new Figs. 1, 2a, b, and 3a, b).

Point 3: The statement "The coordinated iron ion has additional ligands on both sides of FeAPC's plane" is incorrect. Prior oxidation, Fe(II)APC is planar and the absence of binding can't be explained by the presence of additional ligands.

Response 3

We have rewritten the sentence in revised manuscript as follows (page 7, lines 15 – 17):

“Because the coordinated iron ion yields additional ligands at axial positions^{39, 40}, FeAPC stacking is offset, which probably inhibits the π - π interaction between FeAPC and the G-quartet.”

Although further studies are required to elucidate how FeAPC did not bind NRAS RNA, such investigation is out of the scope of this manuscript. Based on the low binding affinity and ROS production, we utilized FeAPC as a negative control for further studies.

Point 4: Binding affinity to G4 motifs have been indirectly assessed using UV-spectroscopy quantifying the ratio between the monomeric/aggregated forms of the phthalocyanine derivatives. The reported dissociation constants are therefore describing the effect of the G4 on this equilibrium and do not solely reflect binding. Fluorescence spectroscopy, using low concentration of the G4s, could be used to this end. This point is crucial in order to understand why ZnAPC is able to degrade NRAS but not BCL2 and VEGF G4s.

Response 4

In accordance with this comment, we evaluated the binding of ZnAPC to G4s by fluorescence titration experiments. In agreement with the results of VIS-spectroscopy, ZnAPC preferentially bound to NRAS RNA than to VEGF RNA or BCL2 RNA. In addition, even with much more points at the lower concentration range of the RNA G-quadruplexes, titration curves traced by VIS absorption and fluorescence are almost identical with each other, and the K_d values evaluated by the VIS absorption and fluorescence titration curves are within the error. These results suggest that the monomerization of ZnAPC is critical for the fluorescence emission and then the photo-cleavage. The results have been shown in new Supplementary Figure 7 and stated on page 18 line 17 – page 20 line 9, of the revised manuscript.

Point 5: It is difficult to understand why the authors extensively discuss the ability of ZnAPC to induce ROS since they clearly demonstrate that ROS formation does not contribute to downregulation of NRAS expression. Nonetheless Figure 2 reports experiments to support that “ROS production by photo-irradiation of ZnAPC is accelerated by its binding to the RNA G-quadruplex in MCF-7 cells”. The reported data are not supporting this conclusion. Control experiments in which the level of the NRAS mRNA is modulated and showing different regime of ROS production are needed to support this conclusion.

Response 5

Thank you for the comment. This comment is very useful to reorganize our manuscript to make it more understandable for the readers of the Chemical Communications. This comment is related to Point 10. Please see Response 10.

Point 6: Figure 3a-b reports the ability of ZnAPC to degrade the NRAS G4. It will be interesting to assess how “clean” is the photolysis reaction by studying longer sequences (full UTR) embedding the G4 motifs.

Response 6

We agree that the photo-cleavage of the full-length of 5' UTR of NRAS mRNA should be confirmed. In accordance with this comment, we examined whether ZnAPC induced the

photolysis of NRAS F-RNA, which is the full-length 5' UTR of NRAS mRNA harbouring the G4 motif (the sequence is shown in Supplementary Fig. 2 of the revised manuscript). The DIG-labelled NRAS F-RNA synthesized by in vitro transcription was used after purification. The band intensity of NRAS F-RNA decreased after photo-irradiation (Fig. 3c). In addition, the photo-cleaved products of NRAS F-RNA were clearly observed (Supplementary Fig. 4c). This result demonstrates that NRAS F-RNA can be photo-cleaved by ZnAPC. The finding is now shown in Figure 3c and Supplementary Figure 4 and is stated on page 11 line 13- page 12 line 2, of the revised manuscript.

Point 7: The authors assess the binding and photolysis properties of ZnAPC on the NRAS DNA G4. The molecule, being polyanionic, is probably mainly localised in the cytoplasm of the cells. Did the authors assess where ZnAPC accumulate in MCF7 cells?

Response 7

We examined the localisation of ZnAPC in MCF-7 cells. It was found that ZnAPC was diffused throughout the cytosol in the MCF-7 cells as the reviewer considered. We have added this result (new Supplementary Figure 5) and description (page 11, lines 5 – 6) in the revised manuscript.

Point 8: While the authors clearly ruled out the role of ROS formation on the downregulation of NRAS expression, the evidences for a direct electron transfer to the G4 motif remain scarce. To demonstrate C2'-hydrogen abstraction on the RNA substrate, a plausible hypothesis to explain the absence of activity on the DNA substrate, one could study the degradation products of the photolysis reaction under aerobic and anaerobic conditions (see Paul et al. JACS 2015, 137, 596-599 for example).

Response 8

Thank you for very important suggestion. To address this comment, we conducted the reaction of photolysis of NRAS RNA under aerobic and anaerobic conditions. As expected, in both cases, NRAS RNA was cleaved after photo-irradiation (Supplementary Fig. 9, revised manuscript). The photo-cleavage under the anaerobic condition supports the direct energy transfer is critical for the photo-cleavage of NRAS RNA by ZnAPC. We have clearly described this finding in the "Discussion" of our revised manuscript (page 22 line 12 – page 23 line 5).

Reviewer: 3

Point 9: The introduction is very short and inadequately referenced (mainly review articles). Of relevance, the role of NRAS in cancer is not described, the role of RNA G-quadruplexes in mRNA translation is not mentioned, and more importantly the literature describing the presence and role of G-quadruplexes in the NRAS 5'UTR (for example Kumari et al, Nat Chem Biol, 2007) is not mentioned.

Similarly, the manuscript should contain a discussion that describe the results in the context of the literature.

Response 9

Thank you very much for pointing out that our introduction part lacks critical references. We have stated what the referee requested and cited relevant publications in the sections “Introduction” (pages 3 line 1 – page 6 line 2) and “Results” (page 13, lines 3-4) of the revised manuscript. Furthermore, we have carefully rewritten the manuscript accordingly and added the section “Discussion” to the revised manuscript (pages 18 - 23).

Point 10: The results in Figure 2 are slightly striking. Figure 2C shows a very modest increase in ROS production in the presence of NRAS RNA in vitro. However, the results in cells presented in Figure 2D show a very strong increase in ROS production following photo-irradiation. This challenges the selectivity of ZnAPC for NRAS RNA as claimed by the authors.

Response 10

We agree with this comment and removed the results regarding ZnAPC-induced ROS production accelerated by its binding to the RNA G-quadruplex, i.e., Figures 2a–c, 5a of the original manuscript. Instead, we have evaluated ROS production by phthalocyanine derivatives in the absence of NRAS RNA. These results are shown in new Figures 2a, b, 5a of the revised manuscript.

Point 11: Overall, and in many places, the authors claim that ZnAPC is selective for NRAS G-quadruplex sequence. However, this claim is only supported by the fact that ZnAPC does not bind two other G-quadruplex forming sequences (BCL-2 and VEGF) or one sequence of double-stranded RNA. There are thousands of putative G-quadruplex forming sequences in the 5'UTR of mRNAs. The data presented are not convincing to claim that ZnAPC is selective for ZnAPC.

Response 11

We agree and changed the description carefully. Furthermore, all the results regarding BCL2 and VEGF RNAs have been moved to Supplementary Figure 7, and we stated the results in the “Discussion” section (page 18 line 17 - page 20 line 9) of our revised manuscript. Moreover, we examined the binding of ZnAPC with five NRAS RNA mutants, NRAS MT1~5 RNAs, to find the preferred RNA-binding sequences of ZnAPC. It was suggested that the number of the G-quartet plane, as well as the flanking sequences at the both sides of the G-quadruplex-forming sequence are important to determine the binding affinity of ZnAPC. Although all of the G-quadruplex sequences are not able to examine, these results indicate that ZnAPC has a moderate binding selectivity for NRAS RNA. These results are now shown in Supplementary Figure 8, and we stated the results in the “Discussion” section (page 20 line 10 - page 22 line 5) of our revised manuscript.

Point 12: The author claim that ZnAPC combined with photo-irradiation could have potential for cancer treatment. It would be useful if the authors could provide more information on the toxicity profile of ZnAPC. They observed that ZnAPC combined with photo-irradiation leads to cancer cell death, but how selective for cancer cell is this?

Response 12

The photosensitizers include phthalocyanines, which possess low cytotoxicity in the dark and preferentially accumulate in tumour tissue. Moreover, PDT typically involves light in a

wavelength range of 600–800 nm to avoid interference by endogenous chromophores. The photosensitizer absorbs light and then relaxes to the first excited singlet state. Next, the singlet state undergoes conversion to the triplet state when it does not go back to the ground state. An electronic energy of the photosensitizer in the triplet state transfers to oxygen, resulting in formation of cytotoxic reactive oxygen species (ROS) such as singlet oxygen ($^1\text{O}_2$) and superoxide (O_2^-). Thus, cancer cells are killed by these photosensitizers in response to light exposure. We have described this notion in the section “Introduction” with new citations in the revised manuscript (page 3, lines 2–13).

Point 13: The gel portions presented in Figure 3 should be presented in full in supplementary material. If the photocleavage is specific, the decrease in NRAS RNA observed in Figure 3A should be balanced by the presence of bands of shorter length corresponding to the cleaved products.

Response 13

This point is related to Point 6. We now show the full gel image in Supplementary Figure 3a of the revised manuscript. Given that the shorter length of NRAS RNA was not observed, we loaded a double amount of NRAS RNA after photo-irradiation for 0 and 120 min onto a 15% denaturing polyacrylamide gel. The band intensity of its full-length and short length was lower and higher, respectively. This result is shown in Supplementary Figure 3b.

<Additional changes>

1. We added the result of real-time PCR analysis of *NRAS* mRNA expression and cell viability into the new Fig. 5f and h of the revised manuscript. These results support our discussion of the possibility that the photo-cleavage activity of ZnAPC towards the RNA G-quadruplex was induced even under the anaerobic condition (page 22 line 10 - page 23 line 5), which is a common characteristic of malignant tumours, and often promotes invasion and metastasis of cancer cells and causes resistance to chemotherapy and radiotherapy.
2. We replaced the original Figure 5f with the new Figure 5g because data in the original one showed the results in the absence of actinomycin D, which terminates transcription. The results shown in the new Figure 5g are obtained in the presence of actinomycin D.
3. We deleted the original Supplementary Figure 2b from the revised manuscript because it was not mentioned in the manuscript. We are sorry about this mistake.

REVIEWERS' COMMENTS:

Reviewer #1 (Remarks to the Author):

In the revised version of their manuscript, Kawauchi et al. have added data in order to address the technical concerns raised during the first submission. While the manuscript is technically sound, it is still difficult to identify major conceptual advances from existing reports. The authors now cite previous works from Xodo and collaborators, but do not specify that the light-mediated (mRNA photocleavage) anticancer activity of G-quadruplex binder have been previously demonstrated in vivo studying both melanoma cells and melanoma tumour-bearing mice.

Reviewer #3 (Remarks to the Author):

The authors have overall answered my concerns. There are still one additional issues:

The major part of the discussion section is results. The text from Lane 311 to 405 should be moved to the results section. The discussion should be extended.

minor comment:

Balasubramanian's name is incorrectly spelled in lane 70.

Lane 141: rephrase "These results are consisted with the previous our study"

Responses to the reviewers' comments

Ms No.: NCOMMS-18-01146A

Title: An anionic phthalocyanine decreases NRAS expression by breaking down its RNA G-quadruplex

Authors: Keiko Kawauchi, Wataru Sugimoto, Takatoshi Yasui, Kohei Murata, Katsuhiko Itoh, Kazuki Takagi, Takaaki Tsuruoka, Hisae Tateishi-Karimata, Kensuke Akamatsu, Naoki Sugimoto, Daisuke Miyoshi

We wish to the reviewers again for their helpful comments. In accordance with their comments, we have made amendments to the manuscript. Please find our point-by-point responses to each of their comments.

Reviewer #1

Point 1: The authors now cite previous works from Xodo and collaborators, but do not specify that the light-mediated (mRNA photocleavage) anticancer activity of G-quadruplex binder have been previously demonstrated in vivo studying both melanoma cells and melanoma tumour-bearing mice.

Response 1

Accordingly, we clearly state about this point in the revised manuscript as the following: Studies using both melanoma cells and melanoma tumour-bearing mice by the group of Luigi Xodo have successfully shown that porphyrin C₁₄H₂₈-alkyl derivative C14, a photosensitizer identified as a G-quadruplex binding compound, induces growth arrest of cancer cells through downregulation of KRAS by targeting the G-quadruplex in its mRNA^{10, 11}. In these articles, it was shown that C14 inhibits translation of KRAS mRNA even without photo-irradiation and induces light-mediated cleavage of KRAS mRNA, which is thought to be associated with its ability to generate ROS (page23, lines 4-11).

Reviewer: 3

Point 2: The major part of the discussion section is results. The text from Lane 311 to 405 should be moved to the results section. The discussion should be extended.

Response 2

Thank you for the comment. The text from Lane 311 to 326 (original manuscript; page17 line 16 – page 18 line 13) was moved to the result section while the text from Lane 326 to 405 (original manuscript) was remained in the discussion section. Because we want to emphasize the main topic of this manuscript that ZnAPC decreases NRAS expression by breaking down its RNA G-quadruplex, we think that that the text regarding the binding affinity of ZnAPC for other RNA G-quadruplexes is shown in the discussion section to discuss a possibility of sequence-selective photo-cleavage of ZnAPC.

In addition, we would like to mention that the number of words in the main text, which should be no more than 5,000 words in the Nature Communications. In the revised manuscript, not the number of words in the main text is 4,982. Because any text we wrote cannot be deleted from our manuscript, it is difficult to extend discussion.

Point 3: minor comment; Balasubramanian's name is incorrectly spelled in lane 70.

Response 3

We sincerely apologize for this misspelled. It has now been corrected.

Point 4: minor comment; Lane 141: rephrase "These results are consisted with the previous our study".

Response 4

We have rewritten the sentence in revised manuscript (page8, lines 13-14).